# CopulaNet: Learning residue co-evolution directly from multiple sequence alignment for protein structure prediction

Fusong Ju [1,2], Jianwei Zhu [3✉], Bin Shao[3], Lupeng Kong[1,2], Tie-Yan Liu[3], Wei-Mou Zheng[2,4] & Dongbo Bu [1,2✉]

Residue co-evolution has become the primary principle for estimating inter-residue distances of a protein, which are crucially important for predicting protein structure. Most existing approaches adopt an indirect strategy, i.e., inferring residue co-evolution based on some hand-crafted features, say, a covariance matrix, calculated from multiple sequence alignment (MSA) of target protein. This indirect strategy, however, cannot fully exploit the information carried by MSA. Here, we report an end-to-end deep neural network, CopulaNet, to estimate residue co-evolution directly from MSA. The key elements of CopulaNet include: (i) an encoder to model context-specific mutation for each residue; (ii) an aggregator to model residue co-evolution, and thereafter estimate inter-residue distances. Using CASP13 (the 13th Critical Assessment of Protein Structure Prediction) target proteins as representatives, we demonstrate that CopulaNet can predict protein structure with improved accuracy and efficiency. This study represents a step toward improved end-to-end prediction of inter-residue distances and protein tertiary structures.

[1] Key Lab of Intelligent Information Processing, State Key Lab of Computer Architecture, Big-data Academy, Institute of Computing Technology, Chinese Academy of Sciences, Beijing, China. [2] University of Chinese Academy of Sciences, Beijing, China. [3] Microsoft Research Asia, Beijing, China. [4] Institute of Theoretical Physics, Chinese Academy of Sciences, Beijing, China. ✉email: jianwzhu@microsoft.com; dbu@ict.ac.cn

Proteins play critical roles in a wide-range of biological processes including catalyzing metabolic reactions, responding to stimuli, and transporting molecules. These biological activities are largely determined by the fine details of protein tertiary structures [1]. Protein structures can be experimentally determined using nuclear magnetic resonance, X-ray crystallography, and cryogenic electron microscopy[2]; however, these technologies are usually difficult and time-consuming and cannot keep pace with the rapid accumulation of protein sequences[3]. An alternative way is protein structure prediction, which predicts structure for a target protein purely from its amino acid sequence. Generally speaking, protein structure prediction approaches are usually more efficient than the experimental technologies for protein structure determination[4,5].

Major progresses have been made during previous years in protein structure prediction and inter-residue contacts/distances have played important roles. Most of the recent protein structure prediction approaches, such as AlphaFold[6] and trRosetta[7], employ roughly the same three-step diagram: i) estimating inter-residue contacts/distances; ii) constructing a potential function based on the estimated contacts/distances; and iii) optimizing the potential function to build a tertiary structure with minimal potential. This diagram has been shown to be successful when the estimated inter-residue contacts/distances are sufficiently accurate.

The state-of-the-art approaches to estimating the inter-residue contacts/distances share the same cornerstones, i.e., constructing multiple sequence alignment (MSA) for a target protein of interest and then performing residue co-evolution analysis on the resulting MSA[8–10]. The underlying rational is that two residues in close spatial proximity always tend to co-evolve; thus, in turn, residue co-evolutions could be exploited to accurately estimate contacts/distances between residues. The co-evolution relationship is usually inferred from residue correlations carried by MSA. This strategy, however, is always hindered by the indirect correlations among residues: the indirect correlations could lead to transitivity in residue spatial proximity and thereafter incorrect estimation of inter-residue contacts/distances. To derive the direct couplings of residues, a variety of direct coupling analysis (DCA) methods have been proposed using precision matrix (the inverse of covariance matrix) or Potts model[11–15]. Currently the DCA technique is widely used for estimating inter-residue contacts/distances, especially combined with deep neural networks for further refinement. For example, both AlphaFold[6] and RaptorX[16] rely on the inter-residue contacts predicted by CCMpred, a DCA-based approach using the Potts model[17].

Although the DCA technique has been shown to be effective in estimating inter-residue contacts/distances, it still suffers from several drawbacks. An outstanding drawback is the considerable information loss after transforming MSAs into hand-crafted features, say covariance matrices. In fact, the DCA technique is founded on the premise that the direct couplings between two residues can be modeled using pairwise statistics such as covariance[9,10]. However, this premise does not always hold. We demonstrated this possibility using two artifactual proteins $P_1$ and $P_2$ as counterexamples (Fig. 1). In protein $P_1$, two residues $R_1$ and $R_2$ are close, whereas in protein $P_2$, they are far from each other. Despite the substantial difference in the constructed MSAs for $P_1$ and $P_2$, the covariance matrices calculated from these MSAs are completely identical, causing the DCA technique to give identical distance estimations for proteins $P_1$ and $P_2$. In fact, for these two MSAs, any statistic of a single residue, or pairwise statistics of two residues, cannot distinguish them. Like the approaches based on covariance matrix, the widely-used Potts model considers these two types of statistics only, and thus also suffers from the limitation illustrated in Fig. 1. Consequently, a more effective way to extract direct couplings between residues from MSAs is highly desirable.

Here, we report an end-to-end deep neural network framework, called CopulaNet, for estimating inter-residue distances. Unlike the existing methods, CopulaNet learns the conditional joint-residue distributions directly from MSAs rather than the hand-crafted features such as covariance matrices. The CopulaNet consists of three key elements, namely, an MSA encoder, a co-evolution aggregator, and a distance estimator. The MSA encoder processes each homologous protein in MSA individually, and embeds each residue to represent its context-specific mutations observed from homologous proteins of the target protein. For any two residues, the aggregator first calculates outer product of their embeddings derived from each homologous protein, then aggregates the outer products acquired from all homologous proteins using average pooling, and finally yields a measure of co-evolution between the two residues. Based on the obtained residue co-evolution, we use a two-dimensional residual network to estimate distance for any residue pairs.

Using CopulaNet as a core module, we develop an approach (called ProFOLD) to protein structure prediction. Briefly speaking, ProFOLD transforms the estimated distances into a potential function, and realizes a tertiary structural conformation with minimal potential. In the following sections, we first demonstrate the concept of ProFOLD using protein T0992-D1 as an example, then apply it to predict structures for the CASP13 target proteins as representatives, and finally compare it with the state-of-the-art prediction approaches. We also present analysis of contributions by the key elements of CopulaNet.

## Results

**Approach summary.** The ProFOLD approach is summarized in Fig. 2. Using the CASP13 target protein T0992-D1 as an example, we demonstrate the concept and main steps of ProFOLD for protein structure prediction.

Protein T0992-D1 consists of a total of 107 residues (only the first 13 residues are shown here for the sake of clear description). For this protein, we first identified its homologs through searching it against protein sequence databases including Uniclust30[18], UniRef90[19] and Metaclust50[20]. The obtained homologous proteins (2807 domains in total) were organized into an MSA. Next, we applied CopulaNet to estimate inter-residue distances directly from the constructed MSA. Here, we inferred the distribution of inter-residue distance over predefined 37 bins, rather than a single distance value. Four examples of these distributions are shown in Supplementary Fig. 1. In the case of residues LEU32 and TYR70, the most likely distance interval was predicted to be (7.5Å, 8Å), which covers the true distance 7.83Å. Finally, we transformed the estimated distance distributions into a potential function, and then searched for the structure conformation with the minimal potential through potential minimization. ProFOLD reports the structural conformation with sufficiently low potential as the final prediction result (shown in the lower-right corner of Fig. 2), which perfectly approximates the native structure (TMscore: 0.84).

The core of our ProFOLD approach is CopulaNet, a deep neural network specially designed to learn inter-residue distances directly from MSA. CopulaNet achieves this objective using three key modules, namely, *MSA encoder*, *co-evolution aggregator*, and *distance estimator*, which are described as below.

*MSA encoder* aims to model the mutations of each individual residue of target protein. Here, we represent an MSA with $K$ homologous proteins as $K$ pairwise alignments, each of which consists of a homologous protein aligned onto the target protein.

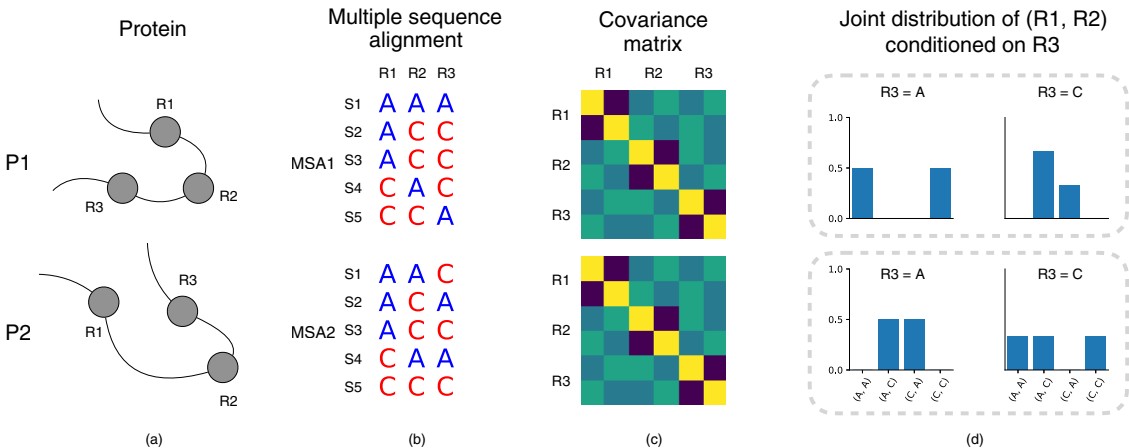

**Fig. 1 The limitation of the covariance-based methods in estimating inter-residue distances. a** Two artifactual proteins $P_1$ and $P_2$. In protein $P_1$, two residues $R_1$ and $R_2$ are close, whereas in protein $P_2$, they are far from each other. **b** The MSAs constructed for the two proteins show considerable difference. **c** The covariance matrices calculated from these two MSAs are totally identical; thus, the covariance-based methods give the same estimation of inter-residue distances for protein $P_1$ and $P_2$. This is contradict the true inter-residue distances. **d** Unlike the covariance matrices, the conditional joint-residue distribution $P(R_1, R_2 | R_3)$ could effectively distinguish these two MSAs.

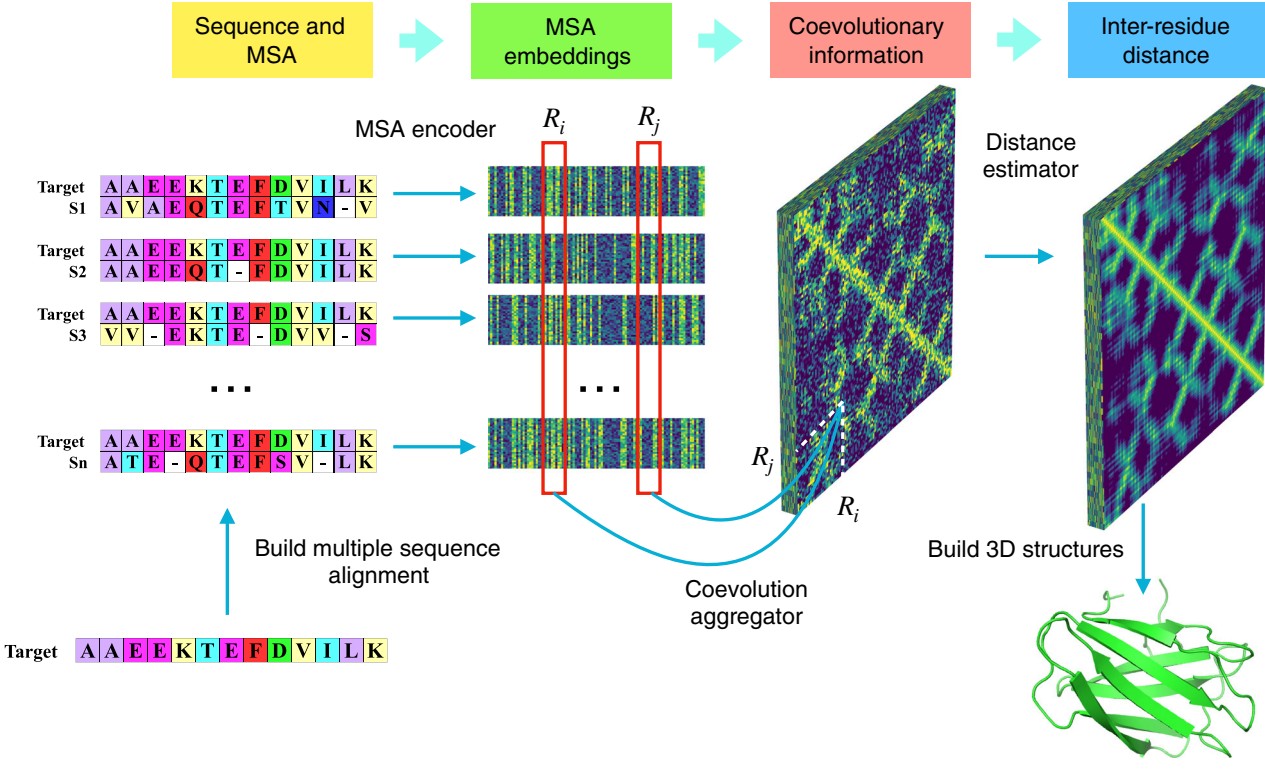

**Fig. 2 Predicting protein tertiary structure using ProFOLD.** Here, we use the CASP13 target protein T0992-D1 as an example to describe the main steps of ProFOLD. Only the first 13 residues are shown here for the sake of clear description. First, we search this protein against sequence databases to identify its homologous proteins (2,807 proteins in total). Next, we use the acquired homologous protein to construct an MSA for this protein. Then we apply CopulaNet to infer residue co-evolution directly from the MSA. CopulaNet uses an MSA encoder to model the mutation information for each residue of the target protein, and then uses a co-evolution aggregator to measure the residue co-mutations. According to the acquired residue co-evolution information, the distance estimator estimates inter-residue distances. Finally, we transform the estimated distance distributions into a potential function, and then search for the structure conformation with the minimal potential. ProFOLD reports the structural conformation with sufficiently low potential as the final prediction result (TMscore: 0.84).

For each individual alignment, MSA encoder identifies the mutations of each residue of the target protein, and embeds the mutations into a vector of 64 features.

As a residue's mutation is highly related to its neighboring residues, the MSA encoder should consider a residue of interest together with its neighbors during embedding. For this end, we design the encoder as a 1D convolutional residual network[21] with multiple convolution layers, thus enabling it to embed a residue together with its neighbors.

*Co-evolution aggregator* measures the co-evolutions between each pair of residues. For any two residues, the aggregator first calculates outer product of their embedding features derived from

a certain homologous protein. As the embedding features of a residue encode its mutations, the outer product of two residues' embedding features could effectively measure the strength of co-mutations between them. Next, by using an average-pooling layer, the aggregator calculates the average outer product obtained from all homologous proteins, thus providing thorough information of co-evolutions between the two residues. Further details of outer product and average-pooling layer are shown in the Methods section and Supplementary Fig. 2.

*Distance estimator* aims to estimate inter-residue distances according to the acquired residue co-evolutions. Previous studies have revealed several structure-related patterns existing in the inter-residue distances. Specifically, two contacting parallel β-strands often form a diagonal line, whereas two contacting anti-parallel β-strands often form an anti-diagonal line. In contrast, two contacting helices usually form a dashed line[22]. Here, we apply a 2D-ResNet to learn these patterns, and thereafter assign these patterns to the estimated inter-residue distances.

To alleviate the difficulty in distance estimation, we transform the distance estimation problem into a classification problem, which is much easier to accomplish. In particular, as performed by trRosetta[7], we divide the inter-residue distance range into 37 intervals, i.e., (0Å, 2.5Å), (2.5Å, 3.0Å), · · · , (19.5Å, 20.0Å), and (20.0Å, + ∞). For each residue pair, CopulaNet predicts a distance distribution over the 37 intervals instead of a single estimated distance value.

### Estimating inter-residue distances using CopulaNet.
Using CopulaNet, ProFOLD estimated inter-residue distances for all 104 CASP13 protein domains. For the sake of fair comparison, we evaluated these estimations in terms of precision of the predicted inter-residue contacts rather than inter-residue distances. Specifically, for two residues, we summed up the predicted probability mass of the intervals with distance below 8Å, and used the sum as predicted probability for the two residue being in contact. As shown in Fig. 3, on the 31 FM domains, ProFOLD achieved prediction precision of 0.840, 0.713 and 0.567 for the most probable $L/5$, $L/2$ and $L$ long-range residue contacts, which is

significantly higher than A7D (AlphaFold), the winner group of CASP13, by 0.128, 0.117 and 0.097, respectively. We also compared with the updated RaptorX[23]. The prediction results of the updated RaptorX were obtained through re-running it using identical MSA to ProFOLD (see Supplementary material for details). As shown in Supplementary Table 1, ProFOLD achieved higher prediction precision than the state-of-the-art approaches.

We further analyzed the contributions of CopulaNet's components for estimating inter-residue distances. As mentioned above, the uniqueness of CopulaNet lies at the use of a learnable "encoder and aggregator" framework rather than traditional statistical models, to infer residue co-evolutions. The obtained residue co-evolutions are further fed into a 2D ResNet to assign the structure-related patterns to inter-residue distances. To examine the contributions by the encoder and aggregator, we built a variant that contains these components only through disabling the 2D ResNet in ProFOLD. The resulting variant, denoted as *ProFOLD w/o R*, was evaluated and compared against the standard ProFOLD.

As shown in Fig. 4, even using the "encoder and aggregator" framework alone, the variant *ProFOLD w/o R* achieved a precision of 0.382 for the top $L$ contact predictions on the CASP13 FM targets. The application of the 2D ResNet in ProFOLD further improved the precision by 0.185. In addition, for both CASP13 FM targets and validation dataset, the performance of *ProFOLD w/o R* increases with the receptive field size, implying that encoding more neighbors surrounding residues will greatly facilitate distance estimating (Fig. 4). In the study, we used an MSA encoder with a receptive field size of 33 (16 1D convolution layers, each layer with a kernel size of 3).

We also implemented three baseline models through replacing the "encoder and aggregator" components of ProFOLD with covariance matrix (a full $L \times L \times 21 \times 21$ matrix) and CCMpred output (also a full $L \times L \times 21 \times 21$ matrix), respectively. The baseline model that trains our 2D ResNet using the CCMpred output together with sequence profile is denoted as baseline-CCM, whereas the baseline model that trains our 2D ResNet using covariance matrix together with sequence profile is denoted as baseline-Cov. We further implemented a baseline model

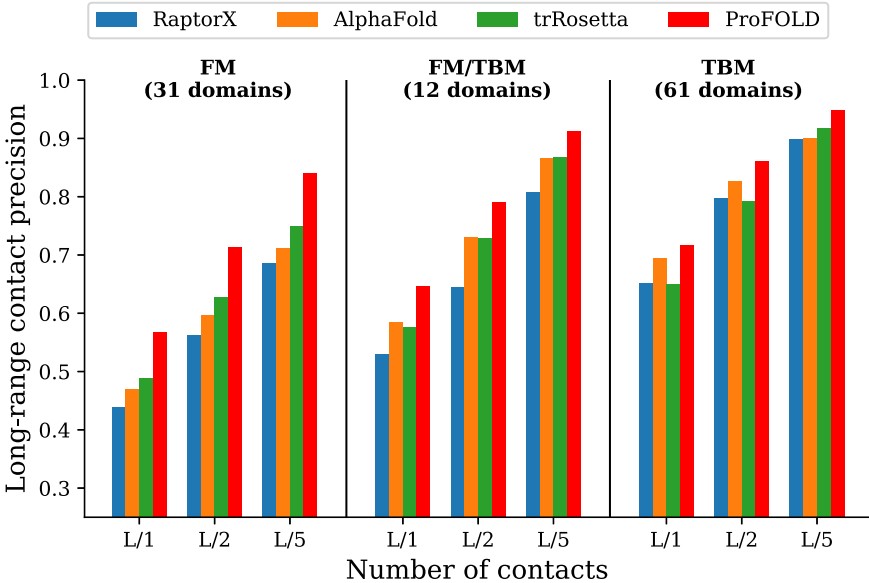

**Fig. 3 Precision of the predicted inter-residue contacts.** Here, the most probable $L/5$, $L/2$ and $L$ long-range residue contacts are shown, where $L$ represents protein length. The phrase "long-range" refers to two residues with sequence separation over 24 residues. For all CASP13 target proteins, ProFOLD outperformed the state-of-the-art approaches. In particular, for the 31 FM domains, ProFOLD achieved precision of 0.840, 0.713 and 0.567 for the most probable $L/5$, $L/2$ and $L$ contacts, which is significantly higher than AlphaFold, by 0.128, 0.117 and 0.097, respectively.

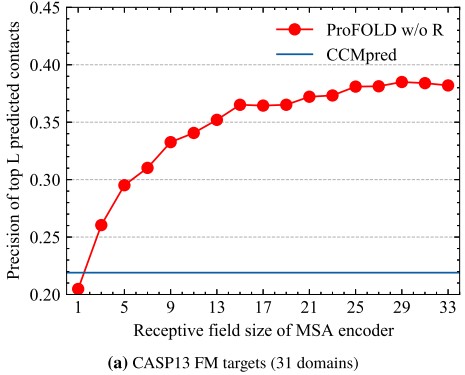
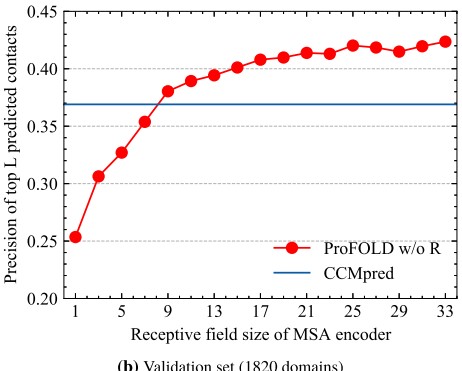

**(a)** CASP13 FM targets (31 domains)

**(b)** Validation set (1820 domains)

**Fig. 4 Precision of the predicted inter-residue contacts by the variant *ProFOLD w/o R*. a** For the 31 CASP13 FM targets, the precision increases with the receptive field size and finally reaches 0.382. **b** On the validation set with 1820 proteins, the precision also increases with the receptive field size and finally reaches 0.424. Even using the "encoder and aggregator" framework alone, the variant *ProFOLD w/o R* still outperformed CCMpred on the two datasets (0.219 and 0.382, respectively).

(called baseline-CF) that uses comprehensive features, including amino acid types, sequence profile, predicted secondary structure, mutual information, covariance matrix and CCMpred output (see Supplementary material for further details).

The performance of these two baseline models is summarized in Supplementary Table 2. As shown in this table, on the 31 CASP13 FM targets, ProFOLD achieved higher precision for long-range contact predictions than the baseline model baseline-CCM (0.567 vs. 0.466, 0.713 vs. 0.603, and 0.840 vs. 0.738 for the most probable $L$, $L/2$ and $L/5$ contacts, respectively) and baseline-Cov (0.567 vs. 0.449, 0.713 vs. 0.591, 0.840 vs. 0.713 for the most probable $L$, $L/2$ and $L/5$ contacts, respectively). Although baseline-CF uses comprehensive features and shows performance improvement, ProFOLD still outperformed baseline-CF (0.567 vs. 0.481, 0.713 vs. 0.621, 0.840 vs. 0.749 for the most probable $L$, $L/2$ and $L/5$ contacts, respectively). The superiority of ProFOLD over these baseline models is also observed on the validation set, even if shallow 2DResNet is used.

Taken together, these results clearly suggested that the main contribution to estimation of inter-residue distances comes from the learnable "encoder and aggregator" framework.

**Predicting protein tertiary structures using ProFOLD.** We applied ProFOLD to predict protein tertiary structures and compared it with the state-of-the-art approaches including AlphaFold (A7D group in CASP13)[6], trRosetta[7], top server groups, and top human groups reported by the CASP13 organizer. The prediction results of AlphaFold, top human groups and top server groups were downloaded from CASP13 official website (https://predictioncenter.org/download_area/CASP13/predictions_trimmed_to_domains/), whereas the prediction results of trRosetta were obtained through re-running it using identical MSA to ProFOLD. The details of these prediction results are summarized in Supplementary Table 3 and Fig. 3.

As shown in Fig. 5a and Supplementary Table 3, on the 31 FM CASP13 datasets, ProFOLD outperformed the state-of-the-art approaches. Specifically, when using the popular cut-off threshold for high-quality structures (TMscore ≥0.70)[6], ProFOLD predicted high-quality structures for 18 out of the 31 domains, whereas AlphaFold and trRosetta predicted high-quality structure for only 12 and 7 domains, respectively. Moreover, the average TMscore of ProFOLD's prediction results is 0.662, which is much higher than that of trRosetta (0.584) and A7D (0.580). Head-to-head comparison clearly demonstrates the advantages of ProFOLD over AlphaFold: for 24 out of the 31 FM domains, ProFOLD

outperformed AlphaFold (Fig. 5b). ProFOLD also outperformed trRosetta on these FM targets (Supplementary Fig. 4).

We also evaluated ProFOLD on the 61 CASP13 TBM and 12 TBM/FM target proteins. For these proteins, although similar template structures are available, ProFOLD predicted their structures in pure ab initio mode without any reference to the template structure information. As shown in Supplementary Table 3, for these targets, the average TMscore of ProFOLD's prediction results is 0.784, which is extremely close to the state-of-the-art template-modeling approach Zhang-server (0.787). These results clearly illustrated that the structural information carried by templates might not be necessary for protein structure prediction. Using the accurate estimation of inter-residue distances by CopulaNet, ProFOLD could predict high-quality protein structures without aids of template structures.

We further examined the possible factors that may affect the successful application of ProFOLD. Previous studies have already shown that the quality of predicted structures is highly related to $M_{eff}$, the number of effective homologous proteins recorded in MSA[14]. As shown in Fig. 6a, the correlation coefficient between the logarithm of $M_{eff}$ and the quality of predicted structures by ProFOLD is as high as 0.69. Therefore, as long as the $M_{eff}$ of a target protein exceeds 20, TMscore of the predicted structure for this protein is expected to be over 0.60 with high confidence. For proteins T0953s2-D3, T0981-D2, T0991-D1, and T0998-D1, ProFOLD could not predict high-quality structures. The reason might be the lack of sufficient homologous proteins: for these proteins, $M_{eff}$ is as small as less than 20. How to improve CopulaNet and ProFOLD to suit the target proteins with only a few homologous proteins remains a future study.

For an approach to protein structure prediction, an interesting and important question is whether we can judge the quality of its prediction results in advance. When the native structure of target protein is already known, we can easily evaluate a predicted structure through comparing it with the native structure; however, thing will become challenging when the native structure is not available. Here, for each structure predicted by ProFOLD, we calculate the average probability of top $L$ predicted contacts (denoted as PPC), and use it as estimated quality of the predicted structure. As shown in Fig. 6b, the correlation efficient between PPC and TMscore of the predicted structures is 0.82. This strong correlation enables us to judge the quality of predicted structure by ProFOLD in advance. Specifically, if a target protein has an estimated PPC over 0.60, the TMscore of the predicted structure by ProFOLD is expected to be over 0.60 with high confidence.

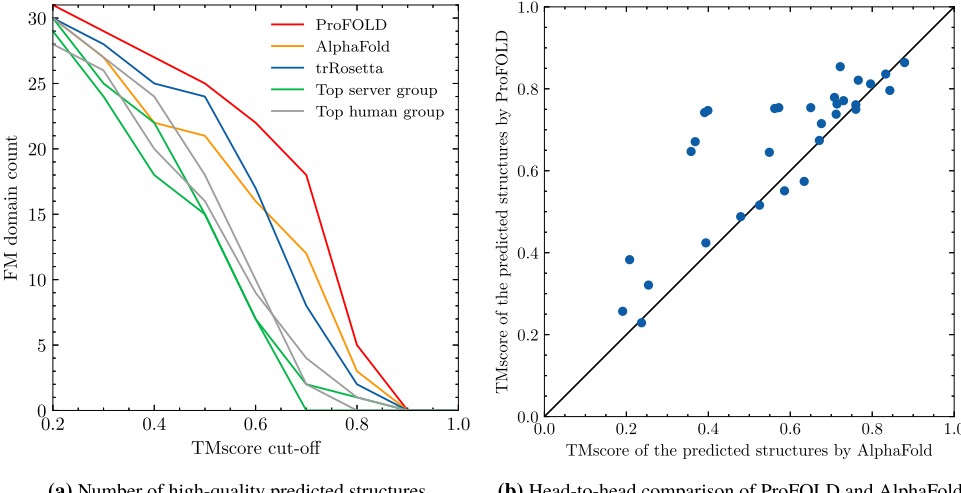

**(a)** Number of high-quality predicted structures      **(b)** Head-to-head comparison of ProFOLD and AlphaFold

**Fig. 5 Quality of the predicted tertiary structures for CASP13 FM target proteins. a** ProFOLD predicted more high-quality structures than the state-of-the-art approaches. When using the popular cut-off threshold for high-quality structures (TMscore ≥0.70), ProFOLD predicted high-quality structures for 18 out of the 31 domains, whereas AlphaFold and trRosetta predicted high-quality structure for only 12 and 7 domains, respectively. **b** Head-to-head comparison clearly demonstrates the advantages of ProFOLD over AlphaFold: for 24 out of the 31 FM domains, ProFOLD outperformed AlphaFold.

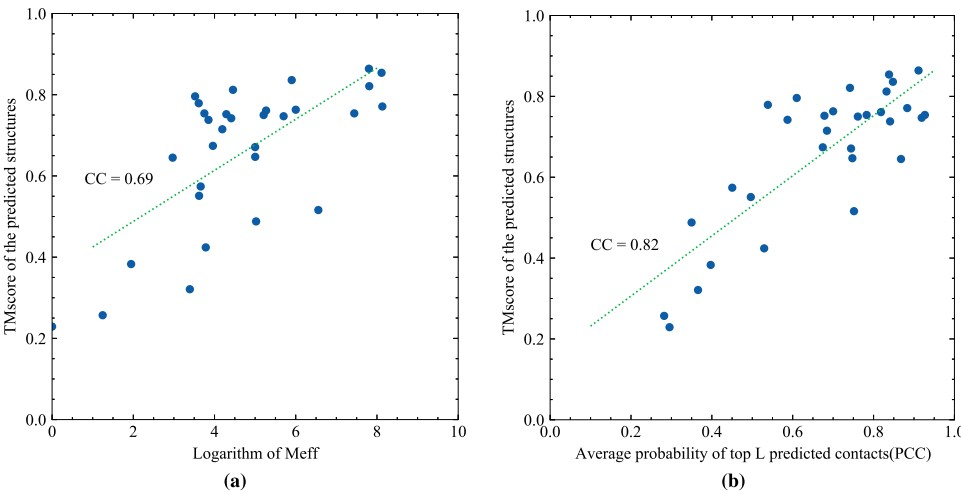

**(a)**      **(b)**

**Fig. 6 Investigation of possible factors that might affect the performance of ProFOLD.** Correlation between quality of the predicted structures and (**a**) $M_{eff}$, (**b**) the average probability of top $L$ predicted contacts (PPC). For the CASP13 FM target proteins, the correlation coefficient between $M_{eff}$ and TMscore of the predicted structures by ProFOLD is as high as 0.69. The correlation efficient between PPC and TMscore of the predicted structures is 0.82.

When applying ProFOLD on a target protein having multiple domains, we can either predict structure for the whole target protein, or predict structure for each domain individually. In the above experiments, we evaluated our approach using MSAs constructed from domain sequences. For more thorough investigations, we have repeated the entire evaluation using MSAs constructed from the whole-target sequences. As shown in Supplementary Table 3, when using the MSA constructed from the domain sequence, both trRosetta and ProFOLD predict better protein structures than using the MSA constructed from the whole-target sequences (0.668 vs. 0.620 for trRosetta, and 0.743 vs. 0.719 for ProFOLD). However, when considering the 31 FM target proteins only, ProFOLD performs slightly better with the MSA constructed from whole-target sequences.

**Contribution analysis of ProFOLD's components**. To better understand the contribution of ProFOLD's components, we built variants of ProFOLD through disabling each component

individually and then compared these variants with the complete ProFOLD approach. In particular, we first disabled the 2D ResNet in distance estimator and thus obtained a variant called *ProFOLD w/o R*. We further disabled the MSA encoder and obtained another variant called *ProFOLD w/o E+R*. For any two residues in target protein, *ProFOLD w/o E+R* captures the correlation between them without consideration of their neighboring residues; thus, it has roughly the same power to the approach that uses covariance matrix for distance estimation.

Using protein T1022s1-D1 as an example, we showed the qualitative comparison of the variants in Fig. 7. When neither MSA encoder nor 2D ResNet is used, the variant *ProFOLD w/o E+R* performed poorly and failed to generate high-quality distance estimations. This result is consistent with the previous observation on the low performance of the covariance-based approaches[11]. When equipped with the MSA encoder module, the variant *ProFOLD w/o R* could generate relatively more accurate distance estimations than *ProFOLD w/o E+R*. Furthermore, as both MSA encoder and 2D ResNet are enabled, the

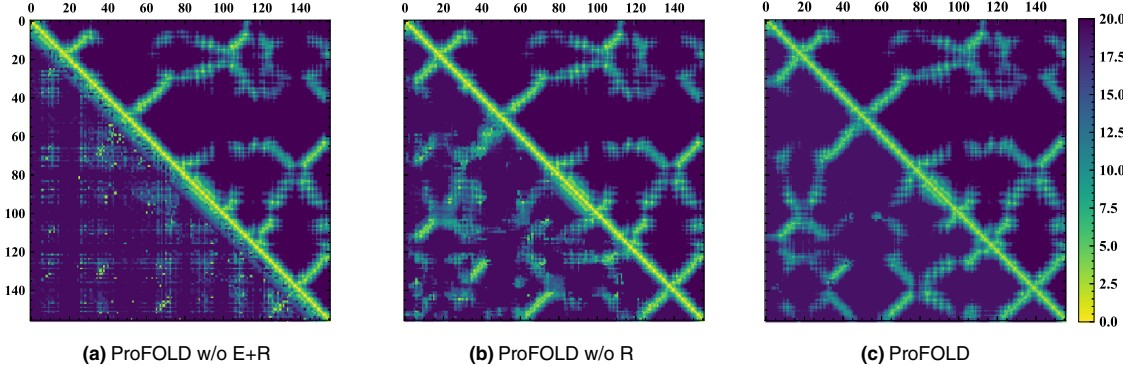

**Fig. 7 Comparison of the predicted inter-residue distances (bottom left) with the ground-truth distances (upper right) for protein T1022s1-D1. a** *ProFOLD w/o E+R* performed poorly and failed to generate high-quality distance estimations. **b** When equipped with the MSA encoder module, the variant *ProFOLD w/o R* could generate relatively accurate distance estimations. **c** When both MSA encoder and 2D ResNet are used, ProFOLD gave distance estimations extremely close to the real distance values.

complete ProFOLD approach gave distance estimations extremely close to the true distance values. These results emphasize the importance of considering neighboring residues in encoding step as well as using the 2D ResNet to learn structure-related patterns existing in inter-residue distances.

To investigate the effect of outer product in co-evolution aggregator, we built another variant of ProFOLD (called *ProFOLD w/o OP*) through disabling the outer product operation. Specifically, we removed the term $g(i, j)$ from equation 2, thus modifying $h(i, j)$ to be the concatenation of $f(i)$ and $f(j)$ only. As shown in Supplementary Figure 5, for the short-range residue contacts (between two residues with sequence separation of 6–11 residues), *ProFOLD w/o OP* showed roughly the same prediction precision as ProFOLD. This is reasonable as the convolution modules in MSA encoder has already effectively modeled the short range relationship. In contrast, for the long-range residue contacts, the prediction precision of *ProFOLD w/o OP* decreased sharply to be significantly lower than ProFOLD. This result clearly demonstrated the importance of the outer product operation in modeling the long-range residue contacts, which cannot be achieved using the convolutional network alone.

**Efficiency of ProFOLD for protein structure prediction**. As described above, ProFOLD learns residue co-evolutions directly from MSA rather than hand-crafted features such as covariance matrix. An MSA might have ten of thousands of homologous proteins, whereas the size of covariance matrix is fixed and determined by the target protein length only. Thus, it is interesting to investigate whether ProFOLD could accomplish protein structure prediction within reasonable time on an average computer.

The three key elements of CopulaNet, i.e., *MSA encoder*, *co-evolution aggregator* and *distance estimator*, exhibit different characteristics in running time and memory requirement. Specifically, unlike the final distance estimator processing 2D information of inter-residue co-evolutions, *MSA encoder* processes 1D sequences only. Moreover, the outer product operations could be efficiently accomplished using the fast matrix multiplication provided by the existing deep neural network frameworks[24]. Thus, compared with distance estimator, both *MSA encoder* and *co-evolution aggregator* modules use only a small amount of computations, making the entire running time insensitive to the number of homologous proteins. Moreover, CopulaNet processes each homologous protein individually; thus, the number of homologous proteins in MSA has little effect on the amount of computer memory required for computing inter-residue distances. ProFOLD also uses MSA sampling and distance

matrix cropping in training process, which could effectively constrain memory usage and avoid potential overfitting as well (see Method section and Supplementary material for further details).

As results, for target proteins with less than 500 residues, ProFOLD could accomplish the whole structure prediction process within 3 hours on an average laptop computer (Intel CPU 2.8G Hz, 16G memory).

**Discussion**
The results presented here for protein structure prediction using ProFOLD have highlighted the special features of learning residue co-evolutions directly from MSA. The abilities of our approach have been clearly demonstrated using CASP13 target proteins as representatives with improved quality of the predicted structures. Using the end-to-end framework CopulaNet, ProFOLD could accurately estimate inter-residue distances and thereafter predict protein structures. The improved efficiency of ProFOLD is an additional advantage, mainly due to the succinct architecture of CopulaNet. It should also be mentioned that the basic idea and architecture of CopulaNet can be readily modified to calculate conditional joint distribution in other fields besides residue co-evolution.

Although in the proof-of-concept study we demonstrated the application of CopulaNet in ab initio prediction of protein structures, the estimated inter-residue distances could also be used to assist template-based prediction approaches. For example, DeepThreader[25] improves threading by incorporating inter-residue distances into scoring function. EigenThreader[26] and CEThreader[27] align target proteins with templates by considering eigenvector decomposition of the predicted inter-residue contacts. These approaches might benefit from the accurate estimation of inter-residue distances provided by CopulaNet.

As CopulaNet attempts to learn residue co-evolution from MSA, it requires that MSA should have sufficient homolog proteins. For the MSAs with only a few homolog proteins, CopulaNet usually cannot accurately estimate inter-residue distances. How to reduce the amount requirement of homolog proteins remains a future study.

Theoretical analysis suggests a possible failure case of our approach. Consider three residues $r_i$, $r_j$, and $r_k$ in the target protein, where both $r_i$ and $r_j$ are in contact with $r_k$ but there is no contact between $r_i$ and $r_j$. If the sequence distance between $r_i$ and $r_k$ (and between $r_j$ and $r_k$) is sufficiently long, MSA encoder cannot perfectly model the effect of $r_k$ on $r_i$ and $r_j$, thus perhaps causing ProFOLD to incorrectly report a contact for residue $r_i$ and $r_j$. The increase of receptive field size in MSA encoder will

partially alleviate this problem; however, when receptive field size is already large, further increase of it will bring limited gains. A perfect way to model long-distance influence among residues is another future study.

In summary, our work on learning residue co-evolution directly from MSA, together with recent developments in constructing high-quality MSAs, will undoubtedly contribute to more accurate prediction of protein tertiary structures and thereafter understanding protein functions.

## Methods

**Architecture of CopulaNet**. CopulaNet consists of three key modules, i.e., *MSA encoder*, *co-evolution aggregator*, and *distance estimator*.

*MSA encoder* embeds residue mutations using a 1D convolutional residual network[21]. The residual network has 8 residual blocks, and each residual block consists of two batch-norm layers, two 1D convolution layers with 64 filters (with kernel size of 3) and exponential linear unit (ELU)[28] nonlinearities (Supplementary Fig. 6).

*Co-evolution aggregator* measures the co-mutations between two residues. Before presenting the design of *co-evolution aggregator* module, we describe the notations to be used first.

Consider a target protein with $L$ residues $t_1 t_2 \cdots t_L$, and a pre-built MSA containing $K$ homologous proteins. By applying *MSA encoder* on the $k$-th homologous protein in MSA, we obtain a total of $C \times L$ embedding features, denoted as $X_k \in \mathbb{R}^{C \times L}$, where $C$ represents the number of output channels of *MSA encoder*. For residue $t_i$ in the target protein, its embedding features extracted from all homologous proteins are aggregated together. The aggregated embedding features, denoted as $f \in \mathbb{R}^{C \times L}$, are calculated as follows.

$$f(i) = \frac{1}{M_{eff}} \sum_{k=1}^{K} w_k X_k(i), \qquad (1)$$

where $w_k$ denotes the weight of the $k$-th homologous protein, and $M_{eff} = \sum_{k=1}^{K} w_k$ represents the sum weight of all homologous proteins. Following the convention established by PSICOV[14], we calculate the weight $w_k$ as the inverse of the number of similar homologous proteins that share at least 80% sequence identity with the $k$-th homolog, and thus $M_{eff}$ represents the number of effective homologous proteins recorded in the MSA.

For two residues $t_i$ and $t_j$ in target protein, the co-evolution aggregator measures their co-mutations using aggregated co-evolution features $h(i, j) \in \mathbb{R}^D$, where $D$ denotes the number of output channels of *co-evolution aggregator* ($D = 4224$ in the study), and $h(i, j)$ refers to the concatenation of the aggregated embedding features and their outer products:

$$h(i, j) = CONCAT(f(i), f(j), g(i, j)). \qquad (2)$$

Here, $g(i, j) \in \mathbb{R}^{C \times C}$ represents the aggregated outer products of the embedding features for residue $t_i$ and $t_j$, which is calculated as below.

$$g(i, j) = \frac{1}{M_{eff}} \sum_{k=1}^{K} w_k [X_k(i) \otimes X_k(j)], \qquad (3)$$

where "$\otimes$" represents the outer product operation.

To summarize, the aggregated co-evolution features consist of $C \times 2$ aggregated embedding features and $C \times C$ aggregated outer product features. In this study, the output channel size $C$ of *MSA encoder* is set as 64. Thus, the co-evolution aggregator generates a total of 4224 ($64 \times 2 + 64 \times 64$) aggregated co-evolution features for any two residues in target protein. An example of the outer product operation is shown in Supplementary Fig. 2 and explained in more details in Supplementary material.

*Distance estimator* aims to estimate inter-residue distances according to the obtained residue co-evolution using a 2D-ResNet with 72 residual blocks. Each block consists of two batch-norm layers, two 2D dilated convolution layers, and exponential linear unit (ELU) nonlinearities.

The further details of the training process are provided in Supplementary material.

**Hyperparameter settings of ProFOLD**. The hyperparameters of ProFOLD were set based on consideration of prediction performance and model size. Specifically, we tested three variants of the 2DResNet used by ProFOLD (72 residual blocks, 96 channels), including "shallow" 2DResNet with only 36 residual blocks, "deeper" 2DResNet with 96 residual blocks, and "wide" 2DResNet with 128 channels.

As illustrated by Supplementary Table 4, the "shallow ProFOLD" shows precision lower than the standard ProFOLD ("shallow ProFOLD": 0.544 vs. standard ProFOLD: 0.567 for the most $L$ probable long-range contacts). Similar observations could be achieved for the two baseline models (Supplementary Table 5). However, when using more channels, the "shallow but wide ProFOLD" shows roughly the same precision as "shallow ProFOLD". These results

demonstrated that the performance of ProFOLD is more sensitive to the number of residual blocks than the number of channels. We also observed that when further increasing the number of residual blocks, the precision is roughly fixed ("deeper ProFOLD": 0.570 vs. standard ProFOLD: 0.567 for the most $L$ probable long-range contacts) but the number of parameters increases sharply.

To balance performance and model size, we used a 2DResNet with 72 residual blocks and 96 channels in the study.

**Benchmark dataset**. In the study, we used the same benchmark dataset as AlphaFold[6]. Briefly speaking, the benchmark dataset was constructed through utilizing 35% sequence similarity cluster representatives of CATH (as of Mar. 16, 2018)[29]. It contains a total of 31,247 non-redundant domains, which was further partitioned into training and validation sets (containing 29,247 and 1,820 proteins, respectively). During the partitioning process, all domains from the same homologous superfamily were allocated to the same partition, thus avoiding potential overlap between partitions.

We tested our methods on CASP13 targets, which consists of 104 domains derived from 71 official targets (the first target was released on May 1, 2018). The 104 domains are officially split into three categories: FM (31 domains), FM/TBM (12 domains) and TBM (61 domains). There is no overlap between training and test sets as the CATH database used in the study were released before testing set.

**MSA generation and representation**. ProFOLD takes multiple sequence alignment of target protein as its only input. For proteins in training and test set, we adopt the same pipeline to construct MSA, i.e., running DeepMSA[30] (with default parameters) against sequence databases Uniclust30 (as of Oct., 2017), UniRef90 (as of Mar., 2018) and Metaclust50 (as of Jan., 2018). All these sequence databases were released before independent test sets and thus there is no overlap between sequence databases and test set.

In the study, we represent the obtained MSA as a collection of sequence pairs. Each sequence pair contains the target protein and a homologous protein. We construct two equal-length strings by adding gaps in aligned sequences so that matching characters are aligned in successive positions (Fig. 2). Then we encode each position as a binary vector of 41 elements, including 20 elements for target protein and 21 elements for homologous protein. Here, the 20 elements for target protein represent 20 amino acid types, and the 21 elements include an extra element to represent gap.

**Structure determination using distance potential**. In the study, we build protein tertiary structures from the predicted inter-residue distances in a similar way to trRosetta[7]. Specifically, we first convert the estimated inter-residue distance distributions into a smooth potential function using the DFIRE[31] paradigm. Then, we use *MinMover* in PyRosetta[32] to search for the tertiary structure with the minimal potential, yielding coarse-grained models with residue centroid only. Finally, these coarse-grained models are refined into full-atom models by executing *FastRelax* in Rosetta.

**Network training setup**. To fit the memory limitation, and as a form of data augmentation, we take a sample of at most 1000 sequences from MSA for a target protein. The largest MSA in the training set consists of a total of 64,780 homologous proteins (for protein 3qhpA). Processing such large MSAs requires large memory, which exceeds the capacity of GPU used in the study. To suit the limited GPU capacity, we randomly extract at most 1,000 homologous proteins as representatives to construct an MSA with appropriate size. In addition, as performed by AlphaFold[6], we also split the distance matrix of a target protein into $128 \times 128$ crops, each of which contains pairwise distances between a group of 128 consecutive residues and another group of 128 consecutive residues. The details of network settings are illustrated in Supplementary materials.

Using both MSA sampling and distance matrix cropping in training process, we could effectively constrain memory usage and avoid potential overfitting as well. In addition, all the training parameters in the proposed neural network are independent of the size of target proteins. Hence, the neural network can handle target proteins and MSAs with arbitrary size during inference.

**Reporting Summary**. Further information on research design is available in the Nature Research Reporting Summary linked to this article.

## Data availability

Our training, validation and test data splits are available via http://protein.ict.ac.cn/ProFOLD. The following versions of public datasets were used in this study: PDB 2018-03; CATH 2018-03; Uniclust30 2017-10; UniRef90 2018-03; and Metaclust 2018-01.

## Code availability

All source codes and models of ProFOLD are publicly available through https://github.com/fusong-ju/ProFOLD. We also developed a web server that is available through http://protein.ict.ac.cn/ProFOLD.

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

## Acknowledgements

We would like to thank the National Key Research and Development Program of China (2018YFC0910405, 2020YFA0907000), and the National Natural Science Foundation of China (31671369, 31770775, 62072435) for providing financial supports for this study and publication charges.

## Author contributions

D.B. directed the protein structure prediction project. F.J., D.B. and J.Z. conceived the study. F.J. designed and implemented the neural network, and performed the computation. F.J., J.Z., B.S., T.L., W.Z., and D.B. analyzed the experimental results. F.J., L.K. and D.B. established the mathematical framework. F.J. and D.B. wrote and revised the manuscript. All authors read and approved the manuscript.

## Competing interests

The authors declare no competing interests.
