## [Peer Review File · Nature Communications]

Reviewers' Comments:

Reviewer #1:

Remarks to the Author:

This manuscript describes a new method for protein structure prediction using deep learning. The major difference from existing methods lies in that this manuscript does not use the classical Potts model (or Markov Random Fields) to model an MSA(multiple sequence alignment). Instead the authors propose to use convolutional neural networks to model each sequence in an MSA, which seems to be a good idea. The authors tested their method using the CASP13 targets and reported better results in both contact prediction and 3D modeling accuracy than existing methods.

Strengths:

- 1) a new idea to deal with MSAs that may work better than traditional methods.
- 2) the reported results seem to be very good.

Major weaknesses:

1) The authors used Fig. 1 to illustrate the weakness of current covariance-matrix-based methods (e.g., PSICOV) in dealing with MSAs. However, the authors did not test their hypothesis illustrated in this Figure. Instead the authors only compared their results with CCMpred, which is not based upon the covariance matrix. Maybe the authors shall compare their method with the same 2D ResNet trained from the covariance matrix. By the way, does CCMpred also suffer from the same issue illustrated in Fig. 1 ?

2) The comparison with CCMpred is not very fair. CCMpred is an unsupervised method but the authors' method is supervised even if 2D ResNet is not used. To do a fair comparison, the authors shall train their 2D ResNet using the CCMpred output (e.g., the full $L \times L \times 21 \times 21$ matrix) as input to see how much better the authors' method is.

3) The results in Supplementary Tables 1 and 2 are inconsistent. For example, in Table 1, the authors showed that on the 12 CASP13 FM/TBM targets, the authors' contact precision is 4-7% higher than trRosetta, but in Table 2, the authors' 3D modeling accuracy is more than 0.11 higher than trRosetta. Since the authors used a similar method as trRosetta to build 3D models from predicted inter-residue distance, it is hard to believe that 4-7% higher contact precision may result in 0.11 higher modeling accuracy. I suggest the authors carefully examine their method in calculating contact precision and 3D modeling accuracy.

Other minor concerns:

1) When predicting contacts and 3D models for CASP13 targets, are the whole-target sequences or the official domain sequences used to construct MSAs?

2) Why the reported contact precision for RaptorX in Supplementary Table 1 is inconsistent with the CASP13 web site?

3) The reported contact precision for trRosetta in Supplementary Table 1 is lower than what's reported by the trRosetta developers. The authors did not explain why. Is it possible due to different strategy of selecting MSAs ? The trRosetta authors tried many different MSAs and then chose the best one, but the authors used their own MSAs as the input of trRosetta.

4) No code or data is found at the provided URL. Is the web server updated to include the method described in this manuscript ?

Questions:

1) What's the largest MSAs in the training data? Does a GPU have enough memory to handle very large MSAs in training?

Reviewer #2:

Remarks to the Author:

The manuscript by Ju et al. is dedicated to developing a new pipeline for protein structure prediction which relies on deep learning techniques. During the last several years we have witnessed how the use of deep learning considerably advanced protein structure prediction with methods like with RaptorX, AlphaFold and trRosetta, among others. Such methods generally follow a similar workflow which consists in taking a multiple sequence alignment for a protein family as input, and predicting inter-residue contacts and/or distances from coevolutionary coupling patterns extracted from the alignment either using pseudolikelihoods, or by covariance matrix inversion, or by making use of plain covariates. Despite all of the above approaches to coevolutionary analysis have proven to provide informative features for the downstream structure prediction networks, they have their inherent limitations of being pairwise, hand-crafted, having limited ability of being fine-tuned in an end-to-end fashion during network training, etc. With the CopulaNet network the authors of the manuscript make a major step forward and show that the above conventional coevolutionary feature extractors can be successfully replaced with a machine-learned one. In the approach suggested by Ju et al. this is done by introducing a multiple sequence encoder and a coevolutionary aggregator which are both inherent parts of the structure prediction network and are learned as a part of it in an end-to-end manner. Inter-residue distance distributions predicted by CopulaNet are then converted to smooth pair restraints and used to reconstruct the 3D structure model by constrained minimization followed by structure relaxation -- pyRosetta was used for both. The full pipeline named ProFOLD was tested on CASP13-derived dataset and shows solid improvement over state-of-the-art methods used in the field. Importance of different constituents of the network were also tested through disabling them individually and comparing to the full ProFOLD pipeline. Overall, this is a high-profile and timely manuscript, which should be of great interest to anyone working on protein structure prediction. Here are a few comments and suggestions how it can be further improved:

1) The main results of the manuscript on the superior performance of the machine-learned coevolutionary extractor are grounded on comparing CopulaNet with the other three deep learning based approaches (RaptorX, AlphaFold, trRosetta) which were all trained on an earlier date, and even despite that, likely relied on somewhat different training sets compared to what was used for CopulaNet. As a result, some of the differences in performance between these methods one could attribute to the differences in the training setups. To address such ambiguities, the authors could train a variant of their network where the MSA encoder and coevolutionary aggregator are replaced with a more traditional coevolutionary extractor (like the MSA covariance matrix or the precision matrix) and use such network as the baseline.

2) From the "Benchmark dataset" section and "MSA generation and representation" section it is not clear which snapshots (release dates) of the PDB and sequence database(s) were used to construct the training set. The authors should clearly indicate the potential overlap between training and test sets both sequence- and structure-wise.

3) More details on the network training setup are worth providing. How many sequences in the MSAs are there? Were full MSAs used for training or were they subsampled? Was any cropping used during training? If it was, at what stage - before or after the MSA encoder? What was the crop size?

4) Is there any evidence for referring to the features which come out of the coevolution aggregator as 'residue co-evolutions' or 'residue co-mutations' vs plain abstract features (p.3)? Are they actually correlated with any couplings which can be obtained from more traditional DCA approaches?

5) There is the entire section on "approximating the conditional joint-residue distribution using CopulaNet" in the Methods, however not much on this topic is reflected in the main text. Is this section needed at all?

6) Low Meff CASP13 targets T0953s2-D3, T0981-D2, T0991-D1, and T0998-D1 mentioned on p.4 all come from bacteriophages. Is it just the low number of homologs in the MSA which affects performance or the type of the organism type also plays the role?

7) I would suggest using 'direct couplings' instead of 'direct correlations' throughout the manuscript.

COMMENTS

Reviewer #1 (Remarks to the Author):

We were pleased that Reviewer #1 approves overall the idea and presentation of our ProFOLD approach and sees it as 'a good idea'. The reviewer also appreciates that our goal of modelling MSA directly using CopulaNet rather than the classical Potts model (or Markov Random Fields). However, this reviewer raised concerns about the model and comparison with other approaches which we address below.

This manuscript describes a new method for protein structure prediction using deep learning. The major difference from existing methods lies in that this manuscript does not use the classical Potts model (or Markov Random Fields) to model an MSA(multiple sequence alignment). Instead the authors propose to use convolutional neural networks to model each sequence in an MSA, which seems to be a good idea. The authors tested their method using the CASP13 targets and reported better results in both contact prediction and 3D modeling accuracy than existing methods.

Strengths:

- 1) a new idea to deal with MSAs that may work better than traditional methods.*
- 2) the reported results seem to be very good.*

We thank this reviewer for his/her overall approval of our work and presentation, and the comments/suggestions to improve the manuscript.

Major weaknesses:

- 1) The authors used Fig. 1 to illustrate the weakness of current covariance-matrix-based methods (e.g., PSICOV) in dealing with MSAs. However, the authors did not test their hypothesis illustrated in this Figure. Instead the authors only compared their results with CCMpred, which is not based upon the covariance matrix. Maybe the authors shall compared their method with the same 2D ResNet trained from the covariance matrix. By*

the way, does CCMpred also suffer from the same issue illustrated in Fig. 1?

We thank the reviewer for pointing out the incomplete test of our hypothesis. We also appreciate the reviewer's suggestion to compare our ProFOLD approach with 'the same 2D ResNet trained from the covariance matrix'. Following this suggestion, we have implemented a new baseline model (denoted as "Covariance matrix+2DResNet") that uses a 2D ResNet trained from the covariance matrix. We have now inserted a new table (Supplementary Table 2) to show the performance of this baseline model and the comparison with ProFOLD.

As illustrated by this new table, on the 31 CASP13 FM targets, ProFOLD achieved higher precision for long-range contact prediction than the baseline model "Covariance matrix+2DResNet" (0.567 vs. 0.449, 0.713 vs. 0.591, 0.840 vs. 0.713 for the most probable L, L/2, L/5 contacts, respectively). The superiority of ProFOLD over the baseline model is also observed on the validation set that contains 1820 domains. These results clearly demonstrated that the combination of MSA encoder and coevolution aggregator is more effective than covariance matrix.

We consider these results are very good and we are pleased to insert these results as a new table (Supplementary Table 2) and a short description (Page 4, Line 106-115, in blue).

In Figure 1, we show two proteins with different MSAs to illustrate the weakness of current covariance-matrix-based methods in distinguishing them. In fact, for these two MSAs, any statistic of a single residue, or pairwise statistics of two residues, cannot distinguish them.

CCMpred attempts to maximize the pseudolikelihood of an L2-regularized Markov random field. It represents columns in an MSA as vertices with single-residue emission potentials and represents covariation between columns as edges with pairwise emission

potentials. Like covariance matrix-based approaches, CCMpred considers statistics of a single residue and pairwise statistics of two residues only, thus also suffering from the limitation illustrated in Figure 1.

2) *The comparison with CCMpred is not very fair. CCMpred is an unsupervised method but the authors' method is supervised even if 2D ResNet is not used. To do a fair comparison, the authors shall train their 2D ResNet using the CCMpred output (e.g., the full $L \times L \times 21 \times 21$ matrix) as input to see how much better the authors' method is.*

We understood the reviewer's concern that 'the comparison with CCMpred is not very fair' since 'CCMpred is an unsupervised method but the authors' method is supervised even if 2D ResNet is not used'. We also appreciate the reviewer for the suggestion of fair comparison. Following this suggestion, we have implemented another baseline model (denoted as "CCMPred+2DResNet") that trains our '2D ResNet using the CCMpred output (the full $L \times L \times 21 \times 21$ matrix)'. The performance of this baseline model, together with that of the "Covariance matrix+2DResNet" baseline model, are listed in the newly-added Supplementary Table 2.

As shown in this new table, on the 31 CASP13 FM targets, ProFOLD achieved higher precision for long-range contact prediction than the baseline model "CCMPred+2DResNet" (0.567 vs. 0.466, 0.713 vs. 0.603, and 0.840 vs. 0.738 for the most probable L, L/2 and L/5 contacts, respectively). Similar observation can be obtained on the validation set.

The comparison of ProFOLD with the baseline models clearly suggested that the main contribution to the estimation of inter-residue distances comes from the learnable "encoder and aggregator" framework.

We have now expanded the description to illustrate this point (Page 4, Line 106-115).

3) The results in Supplementary Tables 1 and 2 are inconsistent. For example, in Table 1, the authors showed that on the 12 CASP13 FM/TBM targets, the authors' contact precision is 4-7% higher than trRosetta, but in Table 2, the authors' 3D modeling accuracy is more than 0.11 higher than trRosetta. Since the authors used a similar method as trRosetta to build 3D models from predicted inter-residue distance, it is hard to believe that 4-7% higher contact precision may result in 0.11 higher modeling accuracy. I suggest the authors carefully examine their method in calculating contact precision and 3D modeling accuracy.

We completely understand the reviewer's concern on the inconsistency between the improvement of contact precision and that of the 3D modelling accuracy.

Following the reviewer's suggestion, we have carefully examined our 'method in calculating contact precision and 3D modelling accuracy'. In the previous submission of our manuscript, we used the script in RaptorX suite (released on 2019.8, downloaded from github <https://github.com/j3xugit/RaptorX-Contact>) to calculate contact precision and used DeepAlign to calculate 3D modelling accuracy. In the revised manuscript, to match the CASP13 assessment, we adopt the CASP13 evaluation metric to calculate contact precision (see the response to minor concern #2 for details). There is a slight difference (~2% for FM targets) between the precisions calculated using these two types of evaluation criteria; however, this does not change the essence of the study.

Now, for the contact predictions by RaptorX and A7D, the calculated contact precision (Supplementary Table 1) and 3D modelling accuracy (Supplementary Table 2 in the previous version, now Table 3 in the revised version) perfectly approximate the official CASP13 assessment, which clearly demonstrates the correctness of calculating contact precision and 3D modelling accuracy.

We think the main reason for the difference between the improvement of contact precision and that of the 3D modelling accuracy might be the small sample size of

FM/TBM targets. Although contact precision strongly correlates with 3D model accuracy, the small sample size (only 12 FM/TBM targets) might lead to considerable perturbation.

For the sake of convenience, we have now uploaded all material, including benchmark datasets, evaluation scripts and evaluation results, onto <http://protein.ict.ac.cn/ProFOLD/ProFOLD.benchmark.tar.gz> for public access.

Other minor concerns:

1) When predicting contacts and 3D models for CASP13 targets, are the whole-target sequences or the official domain sequences used to construct MSAs?

In the first submission of our manuscript, we evaluated our approach using only domain sequences to construct MSAs. For more thorough investigations, we have repeated the entire evaluation using the MSA constructed from the whole-target sequences. We have now expanded Supplementary Table 3 to include these new results.

Generally speaking, when using the MSA constructed from the domain sequence, both trRosetta and ProFOLD predict better protein structures than using the MSA constructed from the whole-target sequences (0.668 vs. 0.620 for trRosetta, and 0.743 vs. 0.719 for ProFOLD). However, when considering the 31 FM target proteins only, ProFOLD performs slightly better with the MSA constructed from whole-target sequences.

We have now added further description to illustrate this point (Page 6, Line 161-166).

2) Why the reported contact precision for RaptorX in Supplementary Table 1 is inconsistent with the CASP13 web site?

We thank the reviewer for pointing out the inconsistency between Supplementary Table 1 and CASP13 website for RaptorX.

The reason of this inconsistency is as follows: In the first submission of our manuscript, we listed in Supplementary Table 1 the contact precision calculated using the script in RaptorX suite. The author of the script, Jinbo Xu, has pointed out that this script usually reports contact precision slightly lower than the official CASP13 assessment as it adopts different definition of protein sequence length (as per personal communications).

To remove the inconsistency and potential misunderstanding, we slightly changed the evaluation procedure:

- i) We re-calculated contact prediction by following the CASP13 official metric rather than running the RaptorX script.
- ii) Following the CASP13 official metric, we also excluded the FM target T0953s1 from evaluation as it has no long-range contacts.

Now, the new evaluation results (listed in the updated Supplementary Table 1) are consistent with the CASP13 official assessment (except for T0953s2, whose domain definition has been changed but the evaluation results have not been changed accordingly). Compared with the results provided in the first submission of this manuscript, the contact precision increased slightly (say, ~2% for FM targets) for all prediction approaches. It should be noted that this slight increase does not change the essence and main conclusion of our manuscript.

In the revised manuscript, we have now updated Supplementary Table 1 and Figure 3.

The original Supplementary Table 1 listed in the first submission and the updated Supplementary Table 1 are shown below for the sake of quick reference.

The original Supplementary Table 1:

Table 1 | Precision of the predicted long-range contacts over CASP13 targets.

Dataset	FM (31 domains)			FM/TBM(12 domains)			TBM (61 domains)		
	L	L/2	L/5	L	L/2	L/5	L	L/2	L/5
RaptorX	0.416	0.536	0.662	0.524	0.643	0.802	0.650	0.797	0.902
A7D	0.448	0.573	0.691	0.579	0.729	0.864	0.692	0.825	0.902
trRosetta	0.466	0.602	0.720	0.568	0.721	0.857	0.648	0.791	0.917
ProFOLD	0.536	0.673	0.808	0.641	0.774	0.899	0.712	0.850	0.940

The updated Supplementary Table 1:

Table 1 | Precision of the predicted long-range contacts over CASP13 targets for the most probable L, L/2 and L/5 contacts. Following the CASP13 criteria, T0953s1 is excluded as it has no long-range contacts

Dataset	FM (31 domains)			FM/TBM (12 domains)			TBM (61 domains)		
	L	L/2	L/5	L	L/2	L/5	L	L/2	L/5
RaptorX	0.439	0.562	0.686	0.530	0.645	0.807	0.652	0.798	0.899
A7D	0.470	0.596	0.712	0.584	0.731	0.866	0.694	0.826	0.901
trRosetta	0.489	0.627	0.750	0.576	0.728	0.867	0.649	0.792	0.917
ProFOLD	0.567	0.713	0.840	0.646	0.791	0.913	0.717	0.860	0.948

3) *The reported contact precision for trRosetta in Supplementary Table 1 is lower than what's reported by the trRosetta developers. The authors did not explain why. Is it possible due to different strategy of selecting MSAs? The trRosetta authors tried many different MSAs and then chose the best one, but the authors used their own MSAs as the input of trRosetta.*

We thank the reviewer for pointing out that *'the reported contact precision for trRosetta in Supplementary Table 1 is lower than what's reported by the trRosetta developers.'*

The source of the difference between our evaluation results and those reported by the trRosetta developers might be as follows:

- i) **Difference in MSAs:** We completely agreed with the reviewer in the point of MSA selection strategy. The trRosetta authors tried multiple MSAs and selected the best one. In contrast, we constructed only one MSA, which was used to evaluate both trRosetta and ProFOLD.
- ii) **Difference in target list:** trRosetta developers performed evaluation on 32 CASP13 target proteins (server group). In contrast, we performed evaluation on 31 CASP13 target proteins (human group) with a server-only target protein excluded.
- iii) **Difference in evaluation criteria:** trRosetta follows CASP13 metric for evaluation. In contrast, we used the RaptorX script for evaluation in the first submission of the manuscript.

As mentioned in the response to minor concern #2, we have now adopted the CASP13 metric instead of RaptorX script for evaluation. As shown in the updated Supplementary Table 1, the precision of long-range contacts prediction for trRosetta increased from 0.466 to 0.489 (top L contacts for FM targets), thus narrowing down the difference with

the precision reported by the trRosetta authors to nearly 2%. This small gap might be due to the difference in MSA selection strategy and the difference in target list as well.

4) *No code or data is found at the provided URL. Is the web server updated to include the method described in this manuscript?*

In the first submission of the manuscript, we have uploaded code and data onto the Nature Communication manuscript center. We did not upload code and data onto the provided URL to avoid potential conflicts with Nature Communication's policy.

We have now updated the web server (<http://protein.ict.ac.cn/ProFOLD/>) to include the method described in this manuscript. We have also uploaded all material, including benchmark datasets, evaluation scripts and evaluation results, onto <http://protein.ict.ac.cn/ProFOLD/ProFOLD.benchmark.tar.gz> for the sake of convenience. We are now developing an automated prediction server using ProFOLD for free access by academic community.

Questions:

1) *What's the largest MSAs in the training data? Does a GPU have enough memory to handle very large MSAs in training?*

The largest MSA in the training data consists of 64,780 homologous proteins (for protein 3qhpA). Processing such large MSAs requires large memory that exceeds the capacity of GPU used in the study. To suit the GPU capacity, as a form of data augmentation, we randomly extracted at most 1000 homologous proteins as representatives to construct an MSA with appropriate size. We also performed distance matrix cropping for reducing memory requirement and improving prediction robustness.

We have now inserted a new subsection "*Network training setup*" to provide more details of network training, including MSA sampling and distance matrix cropping (page 9, line 286-296).

Reviewer #2 (Remarks to the Author):

We were pleased that reviewer #2 approves the idea and presentation of our ProFOLD approach and sees it as 'a major step forward'. The reviewer also appreciates that our goal of modelling MSA directly using CopulaNet towards protein structure prediction 'in an end-to-end manner'. This reviewer also raised several fundamental concerns about the model and evaluations which we address below.

The manuscript by Ju et al. is dedicated to developing a new pipeline for protein structure prediction which relies on deep learning techniques. During the last several years we have witnessed how the use of deep learning considerably advanced protein structure prediction with methods like with RaptorX, AlphaFold and trRosetta, among others. Such methods generally follow a similar workflow which consists in taking a multiple sequence alignment for a protein family as input, and predicting inter-residue contacts and/or distances from coevolutionary coupling patterns extracted from the alignment either using pseudolikelihoods, or by covariance matrix inversion, or by making use of plain covariates. Despite all of the above approaches to coevolutionary analysis have proven to provide informative features for the downstream structure prediction networks, they have their inherent limitations of being pairwise, hand-crafted, having limited ability of being fine-tuned in an end-to-end fashion during network training, etc. With the CopulaNet network the authors of the manuscript make a major step forward and show that the above conventional coevolutionary feature extractors can be successfully replaced with a machine-learned one. In the approach suggested by Ju et al. this is done by introducing a multiple sequence encoder and a coevolutionary aggregator which are both inherent parts of the structure prediction network and are learned as a part of it in an end-to-end manner. Inter-residue distance distributions predicted by CopulaNet are then converted to smooth pair restraints and used to reconstruct the 3D structure model by constrained minimization followed by structure relaxation -- pyRosetta was used for both. The full pipeline named ProFOLD was tested on CASP13-derived dataset and shows solid improvement over state-of-the-art methods used in the field. Importance of different constituents of the network

were also tested through disabling them individually and comparing to the full ProFOLD pipeline. Overall, this is a high-profile and timely manuscript, which should be of great interest to anyone working on protein structure prediction. Here are a few comments and suggestions how it can be further improved:

We appreciate this reviewer for his/her invaluable comments/suggestions to improve the manuscript.

1) The main results of the manuscript on the superior performance of the machine-learned coevolutionary extractor are grounded on comparing CopulaNet with the other three deep learning based approaches (RaptorX, AlphaFold, trRosetta) which were all trained on an earlier date, and even despite that, likely relied on somewhat different training sets compared to what was used for CopulaNet. As a result, some of the differences in performance between these methods one could attribute to the differences in the training setups. To address such ambiguities, the authors could train a variant of their network where the MSA encoder and coevolutionary aggregator are replaced with a more traditional coevolutionary extractor (like the MSA covariance matrix or the precision matrix) and use such network as the baseline.

We thank the reviewer for pointing out the ambiguities caused by insufficient comparison. We also appreciate the reviewer for the suggestions to train a variant of our network 'where the MSA encoder and coevolutionary aggregator are replaced with a more traditional coevolutionary extractor (like the MSA covariance matrix or the precision matrix)'. Following this suggestion, we have implemented two baseline models:

- i) Baseline model "Covariance matrix+2DResNet": In this baseline model, we replaced the MSA encoder and coevolutionary aggregator with covariance matrix, i.e., training our 2D ResNet from the covariance matrix.
- ii) Baseline model "CCMPred+2DResNet": In this baseline model, we replaced the MSA encoder and coevolutionary aggregator with CCMpred, i.e.,

training our 2D ResNet using the CCMPred output (the full $L \times L \times 21 \times 21$ matrix).

We have now inserted a new table (Supplementary Table 2) to show the performance of these baseline models and the comparison with ProFOLD.

As shown in this new table, on the 31 CASP13 FM targets, ProFOLD achieved higher precision for long-range contact prediction than the baseline model "CCMPred+2DResNet" (0.567 vs. 0.466, 0.713 vs. 0.603, and 0.840 vs. 0.738 for the most probable L, L/2 and L/5 contacts, respectively) and the baseline model "Covariance matrix+2DResNet" (0.567 vs. 0.449, 0.713 vs. 0.591, 0.840 vs. 0.713 for the most probable L, L/2, L/5 contacts, respectively). The superiority of ProFOLD over these baseline models is also observed on the validation set that contains 1820 domains.

Taken together, the comparison of ProFOLD with these two baseline models clearly demonstrated that the main contribution to the estimation of inter-residue distances comes from the learnable "encoder and aggregator" framework.

We consider these results are very good and we are pleased to insert these results as a new table (Supplementary Table 2) and a short description (Page 4, Line 106-115, in blue).

2) From the "Benchmark dataset" section and "MSA generation and representation" section it is not clear which snapshots (release dates) of the PDB and sequence database(s) were used to construct the training set. The authors should clearly indicate the potential overlap between training and test sets both sequence- and structure-wise.

In the study, for the sake of fair comparison, we used the benchmark dataset identical to AlphaFold. Briefly speaking, the benchmark dataset was constructed through utilizing the CATH 35% sequence similarity cluster representatives. It contains a total of 31,247 non-

redundant domains, which was further partitioned into training and validation sets (containing 29,247 and 1,820 proteins, respectively). To avoid potential overlap, during the data partitioning process, all domains from the same homologous superfamily were kept in the same partition.

For proteins in both training and test set, we adopt the same pipeline to construct MSA, i.e., running DeepMSA (with default parameters) against sequence databases Uniclust30 (as of Oct., 2017), UniRef90 (as of Mar., 2018) and Metaclust50 (as of Jan., 2018). All these sequence databases were released before independent test sets and thus there is no overlap between sequence databases and test set.

We have now revised the description to clearly show the release dates of PDB, CATH, and sequence databases (page 8, line 260-274). We have also expanded the description to indicate the efforts to reduce the overlap between training and test sets (page 8, line 260-265).

3) More details on the network training setup are worth providing. How many sequences in the MSAs are there? Were full MSAs used for training or were they subsampled? Was any cropping used during training? If it was, at what stage - before or after the MSA encoder? What was the crop size?

We thank the reviewer for pointing out this issue. We realized that we did not provide sufficient details about network training setup. We have now inserted a new subsection “*Network training setup*” to provide the details of network training, including MSA sampling and distance matrix cropping.

Briefly speaking, the largest MSA in the training data consists of 64,780 homologous proteins (for protein 3qhpA). Processing such large MSAs requires large memory, which exceeds the capacity of GPU used in the study. To suit the GPU capacity, and as a form of data augmentation, we randomly sampled 1000 homologous proteins as representatives to construct an MSA with appropriate size. In addition, as performed by AlphaFold, we

also split the distance matrix of a target protein into 128x128 crops, each of which contains pairwise distances between a group of 128 consecutive residues and another group of 128 consecutive residues.

In the study, we performed cropping before MSA encoder. We appreciate the reviewer for pointing out the possibility of cropping after MSA encoder. We will investigate this operation in future study.

Using MSA sampling and distance matrix cropping in training process, we could effectively constrain memory usage and avoid potential overfitting as well. In addition, all the training parameters in the proposed neural network are independent of the size of target proteins. Hence, the neural network can handle target proteins and MSAs with arbitrary size during inference.

4) Is there any evidence for referring to the features which come out of the coevolution aggregator as 'residue co-evolutions' or 'residue co-mutations' vs plain abstract features (p.3)? Are they actually correlated with any couplings which can be obtained from more traditional DCA approaches?

We appreciate the reviewer for raising the point on interpreting the network, especially the coevolution aggregator's output. Our analysis suggests the following points:

- i) Consider the case that MSA encoder is disabled first. In this case, according to equation (3), $g(i, j)$ calculates the frequency of co-mutation between residue i and j . Thus, theoretically, the features come out of the coevolution aggregator carry information of residue co-mutations.
- ii) The traditional DCA approaches extract co-mutations between residue pairs; thus, in principle, the features come out of the coevolution aggregator are correlated with the output of DCA approaches. However, our approach utilizes MSA encoder to extract context-specific information

related to residue i and j , thus carry more information than the traditional DCA approaches.

5) There is the entire section on "approximating the conditional joint-residue distribution using CopulaNet" in the Methods, however not much on this topic is reflected in the main text. Is this section needed at all?

We thank the reviewer for pointing out the weak connection of this section to the main text. In the first submission of the manuscript, the section was added to describe a possible interpretation of outer production operations. We realized that the connection of this section to the main text is weak.

Following the reviewer's suggestion, we have now removed this section from the main text of the manuscript and put it in Supplementary material with simplification.

6) Low Meff CASP13 targets T0953s2-D3, T0981-D2, T0991-D1, and T0998-D1 mentioned on p.4 all come from bacteriophages. Is it just the low number of homologs in the MSA which affects performance or the type of the organism type also plays the role?

In the first submission of our manuscript, we constructed MSAs for these four targets through running DeepMSA. The constructed MSAs have low Meff for these targets (T0953s2-D3, T0981-D2, T0991-D1, and T0998-D1, respectively), leading to low-quality predictions by our approach.

In-depth examination illustrated that this failure can be partly due to the inappropriate filtering criteria (default setting: $e\text{-value}=0.001$) used by DeepMSA. By manually increasing $e\text{-value}$ to be 1, we could identify more homologous proteins. In the case of target T0981-D2, the quality of the final predicted structure increased from 0.257 to 0.586.

These results clearly demonstrated that when sufficient homologous proteins are provided, our approach could make accurate predictions for the proteins from bacteriophages. Searching against larger meta-genome sequence databases would help to improve MSAs for the proteins from bacteriophages.

7) I would suggest using 'direct couplings' instead of 'direct correlations' throughout the manuscript.

This has been changed as suggested.

Reviewers' Comments:

Reviewer #1:

Remarks to the Author:

I appreciate that the authors did more experiments to support their claims in the manuscript. However, I am afraid that the authors did not do the baseline experiments in an effective way. The accuracy of the two baseline methods "CCMPred+2DResNet" and "Covariance matrix+2DResNet" (especially CCMpred+2DResNet) obtained by the authors is kind of low compared to what's reported in the literature. This makes it hard to judge the real advantage of the proposed method over existing methods.

Reviewer #2:

Remarks to the Author:

I thank the authors for clearly addressing all my comments.

COMMENTS

Reviewer #1 (Remarks to the Author):

We were pleased that Reviewer #1 approves the further experiments in the revised manuscript. However, this reviewer raised concerns about the performance of the baseline which we address below.

I appreciate that the authors did more experiments to support their claims in the manuscript. However, I am afraid that the authors did not do the baseline experiments in an effective way. The accuracy of the two baseline methods "CCMPred+2DResNet" and "Covariance matrix+2DResNet" (especially CCMpred+2DResNet) obtained by the authors is kind of low compared to what's reported in the literature. This makes it hard to judge the real advantage of the proposed method over existing methods.

We understand the reviewer's concern on performing 'the baseline experiments in an effective way'. The reviewer also pointed out that 'the accuracy of the two baseline methods' is 'kind of low compared with what's reported in the literature'. We realized that in the previous submission, the comparison with the existing methods in literature is not elaborate, making 'it hard to judge the real advantage of the proposed method over existing methods'.

To address this issue, we performed elaborate literature search, and summarized the existing methods in literature as follows.

Method	Dataset	Source of evaluation results	Precision of contact prediction		
			L	$L/2$	$L/5$
RaptorX	CASP11 targets	Ref. Wang2017	0.550	0.680	0.770
RaptorX	CASP13 FM targets	Ref. Senior2020	0.431	0.549	0.673
RaptorX	CASP13 FM targets	in-house evaluation	0.439	0.562	0.686
A7D	CASP13 FM targets	Ref. Senior2020	0.461	0.585	0.699
A7D	CASP13 FM targets	in-house evaluation	0.470	0.596	0.712
CCMpred+2DResNet	CASP13 FM targets	in-house evaluation	0.466	0.603	0.738
trRosetta	CASP13 FM targets	in-house evaluation	0.489	0.627	0.750
Covariance Matrix+2DResNet	CASP13 FM targets	in-house evaluation	0.449	0.591	0.713

The above table suggests the following points.

(1) Comparison of "CCMpred+2DResNet" with A7D and RaptorX:

First, no matter for RaptorX or A7D, our in-house evaluation results perfectly approximate the performance reported in literature. The difference, which is less than 2%, comes from the different evaluation metrics used by the studies. This result clearly demonstrates the correctness of calculating contact precision.

Second, on the CASP13 FM targets, the baseline model "CCMpred+2DResNet" shows contact prediction precision comparable with A7D and higher than RaptorX. For example, "CCMpred+2DResNet", A7D and RaptorX show precision of 0.466, 0.470 and 0.439, respectively, for the most probable L long-range contacts.

Third, we also noticed that the precision of "CCMpred+2DResNet" on CASP13 FM targets is lower than that of RaptorX on CASP11 targets (0.466 vs. 0.550, 0.603 vs. 0.680, 0.738 vs. 0.770 for the most probable L , $L/2$, $L/5$ contacts, respectively). The authors of RaptorX themselves also reported that the performance of RaptorX on CASP13 FM targets (in Ref. Xu2020) is lower than that on CASP11 targets (in Ref. Wang2017); thus, this performance difference should come from the different protein targets in the two test sets rather than the prediction approaches.

(2) Comparison of "Covariance matrix+2DResNet" with trRosetta:

On the CASP13 FM targets, the baseline model "Covariance matrix+2DResNet" showed prediction precision lower than trRosetta (0.449 vs. 0.489, 0.591 vs. 0.627, 0.713 vs. 0.750 for the most probable L , $L/2$, $L/5$ contacts, respectively). This performance difference comes from the fact that the baseline model "Covariance matrix+2DResNet" uses "covariance" of residues derived from query MSA, whereas trRosetta uses the precision matrix, i.e., the inverse of the covariance matrix. The recent study ResPRE (Ref. Li2020) has reported similar observations.

We also realized that the implementation of the two baseline models was not elaborately described in the previous submission. In this revision, we have now added a new section (Section 4 in Supplementary text) to describe the implementation of baseline models, including features, datasets, and neural network architecture.

In summary, the comparison of ProFOLD and the two baseline models clearly demonstrated that the advantages of ProFOLD come from its “end-to-end” framework: using this framework, ProFOLD could learn inter-residue distances directly from MSA and thus avoids the drawbacks of the hand-crafted features.

Besides comparing ProFOLD with baseline models, we carried out an extra experiment to investigate the effects of hyperparameters of 2DResNet. Specifically, we tested three variants of the 2DResNet used by ProFOLD (with 72 residual blocks, 96 channels), including “shallow” 2DResNet with only 36 residual blocks, “deeper” 2DResNet with 96 residual blocks, and “wide” 2DResNet with 128 channels.

As shown in the newly-added Supplementary Table 4, the “shallow ProFOLD” shows precision lower than the standard ProFOLD (“shallow ProFOLD”: 0.544 vs. ProFOLD: 0.567 for the most L probable contacts). Similar observations could be achieved for the two baseline models (extended Supplementary Table 2). However, when using more channels, the “shallow but wide ProFOLD” shows roughly the same precision as “shallow ProFOLD”. These results demonstrated that the performance of ProFOLD is more sensitive to the number of residual blocks than the number of channels. We also observed that when further increasing the number of residual blocks, the precision is roughly fixed (“deeper ProFOLD”: 0.570 vs. ProFOLD: 0.567 for the most L probable contacts) but the number of parameters increases sharply.

To balance performance and model size, we used a 2DResNet with 72 residual blocks and 96 channels in the study.

We consider these results are meaningful and we are pleased to insert these results as a new table (Supplementary tables 4), an extended table (Supplementary Table 2), and a new subsection in Methods (page 8-9, line 269-280).

The extended Supplementary Table 2, the newly-added Supplementary Table 4, and references are listed below for the convenience of reference.

Table 2 | Comparison of ProFOLD and two baseline models in terms of precision of contact predictions. Here, the precision for the most probable L , $L/2$ and $L/5$ long-range contacts over CASP13 targets and validation set are shown. The "shallow" 2DResNet has a total of 36 residual blocks whereas 2DResNet in ProFOLD has a total of 72 residual blocks. The label "CCMPred+2DResNet" denotes the baseline model that trains our 2D ResNet using the CCMPred output, whereas "Covariance matrix+2DResNet" represents the baseline model that trains our 2D ResNet using covariance matrix. Following the CASP13 criteria, T0953s1 is excluded as it has no long-range contacts

Method	CASP13 FM (31 domains)			Validation set (1820 domains)		
	L	$L/2$	$L/5$	L	$L/2$	$L/5$
CCMPred+2DResNet	0.466	0.603	0.738	0.582	0.764	0.871
Covariance Matrix+2DResNet	0.449	0.591	0.713	0.556	0.735	0.852
ProFOLD	0.567	0.713	0.840	0.641	0.831	0.918
CCMPred+shallow 2DResNet	0.445	0.582	0.702	0.569	0.754	0.857
Covariance Matrix+shallow 2DResNet	0.424	0.570	0.687	0.541	0.722	0.838
Shallow ProFOLD	0.544	0.689	0.808	0.632	0.819	0.910

Table 4 | The performance of ProFOLD under various settings of hyper-parameters. Here, the precision for the most probable L , $L/2$ and $L/5$ long-range contacts over CASP13 FM targets and validation set are shown. Following the CASP13 criteria, T0953s1 is excluded as it has no long-range contacts

Method	#Residual			CASP13 FM (31 domains)			Validation set (1820 domains)		
	blocks	#Channels	#Parameters	L	$L/2$	$L/5$	L	$L/2$	$L/5$
Shallow ProFOLD	36	96	6.46 M	0.544	0.689	0.808	0.632	0.819	0.910
Shallow but wide ProFOLD	36	128	11.19 M	0.548	0.688	0.811	0.635	0.821	0.909
ProFOLD	72	96	12.44 M	0.567	0.713	0.840	0.641	0.831	0.918
Deeper ProFOLD	96	96	16.43 M	0.570	0.714	0.839	0.643	0.833	0.922

References

1. S. Wang et al. Accurate de novo prediction of protein contact map by ultra-deep learning model. *PLoS Computational Biology* **13** no. 1, p. e1005324 (2017).
2. J. Xu et al. Improved protein structure prediction by deep learning irrespective of co-evolution information. *bioRxiv* (2020).
3. A. W. Senior et al. Improved protein structure prediction using potentials from deep learning. *Nature* **577**, pp. 706–710 (2020).

Reviewers' Comments:

Reviewer #1:

Remarks to the Author:

I appreciate that the authors did more experiments. I am fine with the response, but it is still unclear to me how much improvement can be obtained by the new component of CopulaNet, i.e., the 1D convolutional network for MSA processing.

It is important to understand this because this is the only new idea introduced by this manuscript.

To do a fair comparison, in addition to using the same 2D ResNet (i.e., the same number of 2D Conv Layers and a similar number of filters/layer) and the same MSAs, the baseline methods shall not ignore sequence profile, mutual information and the full CCMpred co-evolution matrix (of dimension 21L*21L) since the 1D conv network used by the authors may automatically learn these information. That is, I would like to understand how much extra information may be learned by the authors' 1D conv network from MSAs over all manually-engineered features derived from MSAs.

In the letter, the authors said "The authors of RaptorX themselves also reported that the performance of

RaptorX on CASP13 FM targets (in Ref. Xu2020) is lower than that on CASP11 targets (in Ref. Wang2017)". What does this mean? First, it is not very meaningful to compare the performance of one method on two different test sets. Second, RaptorX's performance on the CASP13 FM targets reported in Xu2020 indeed is higher than RaptorX's performance on the whole CASP11 target set reported in Wang2017. By the way, the authors mentioned Xu2020, but did not cite the most recent results reported in this reference.

COMMENTS

Reviewer #1 (Remarks to the Author):

I appreciate that the authors did more experiments. I am fine with the response, but it is still unclear to me how much improvement can be obtained by the new component of CopulaNet, i.e., the 1D convolutional network for MSA processing. It is important to understand this because this is the only new idea introduced by this manuscript.

To do a fair comparison, in addition to using the same 2D ResNet (i.e., the same number of 2D Conv Layers and a similar number of filters/layer) and the same MSAs, the baseline methods shall not ignore sequence profile, mutual information and the full CCMpred co-evolution matrix (of dimension $21L*21L$) since the 1D conv network used by the authors may automatically learn these information. That is, I would like to understand how much extra information may be learned by the authors' 1D conv network from MSAs over all manually-engineered features derived from MSAs.

We understand the reviewer's concern on '*how much improvement can be obtained by the new component of CopulaNet*'. We also greatly appreciate the reviewer for his/her detailed suggestion on implementing an effective baseline model. Following this suggestion, we have implemented a new baseline model (denoted as "*baseline-CF*") that uses '*all manually-engineered features*', including:

- 1D features: one-hot amino acid type, sequence profile, and secondary structure (predicted by PSIPRED).
- 2D features: mutual information, covariance matrix (of dimension $21L*21L$), and full CCMpred co-evolution matrix (of dimension $21L*21L$).

We have extended Supplementary Table 2 to include the performance of baseline-CF together with these of the previous baseline models, i.e., baseline-CCM that uses CCMpred output (of dimension $21L*21L$) and sequence profile as input features, and baseline-Cov that uses covariance matrix (of dimension $21L*21L$) and sequence profile as input features.

Table 2 | Comparison of ProFOLD and baseline models in terms of precision of contact predictions. Here, the precision for the most probable L , $L/2$ and $L/5$ long-range contacts over CASP13 targets and validation set are shown. The “shallow” 2DResNet has a total of 36 residual blocks whereas 2DResNet in ProFOLD has a total of 72 residual blocks. The baseline model baseline-CCM trains our 2D ResNet using CCMpred output (a $L \times L \times 21 \times 21$ matrix) together with sequence profile as input features, whereas baseline-Cov trains our 2D ResNet using covariance matrix (also a $L \times L \times 21 \times 21$ matrix) together with sequence profile as input features. The baseline model baseline-CF uses comprehensive features, including amino acid types, sequence profile, predicted secondary structure, mutual information, covariance matrix and CCMpred output. Following the CASP13 criteria, T0953s1 is excluded as it has no long-range contacts

Method	CASP13 FM (31 domains)			Validation set (1820 domains)		
	L	$L/2$	$L/5$	L	$L/2$	$L/5$
baseline-CCM	0.466	0.603	0.738	0.582	0.764	0.871
baseline-Cov	0.449	0.591	0.713	0.556	0.735	0.852
baseline-CF	0.481	0.621	0.749	0.595	0.778	0.879
ProFOLD	0.567	0.713	0.840	0.641	0.831	0.918
Shallow baseline-CCM	0.445	0.582	0.702	0.569	0.754	0.857
Shallow baseline-Cov	0.424	0.570	0.687	0.541	0.722	0.838
Shallow baseline-CF	0.458	0.595	0.712	0.572	0.763	0.869
Shallow ProFOLD	0.544	0.689	0.808	0.632	0.819	0.910

As shown in the table, on the 31 CASP13 FM targets, ProFOLD achieved higher precision for long-range contact prediction than the baseline model baseline-CF (0.567 vs. 0.481, 0.713 vs. 0.621, and 0.840 vs. 0.749 for the most probable L , $L/2$, and $L/5$ contacts, respectively). Similar observations can be obtained on the validation set. These results clearly show the advantage of machine-learned features over all manually-engineered features derived from MSAs.

We considered these results are meaningful and we were pleased to insert these results as an extended table (Supplementary Table 2). We also extended description in main text (Page 4, Line 114-128) and Supplementary Material.

In the letter, the authors said "The authors of RaptorX themselves also reported that the performance of RaptorX on CASP13 FM targets (in Ref. Xu2020) is lower than that on CASP11 targets (in Ref. Wang2017)". What does this mean? First, it is not very meaningful to compare the performance of one method on two different test sets. Second, RaptorX's performance on the CASP13 FM targets reported in Xu2020 indeed is higher than RaptorX's performance on the whole CASP11 target set reported in Wang2017. By the way, the authors mentioned Xu2020, but did not cite the most recent results reported in this reference.

We completely agree with the reviewer on the issue that “*it is not very meaningful to compare the performance of one method on two different test sets*”. When preparing the second

revision, we misunderstood the phrase “*what’s reported in the literature*” and thus simply listed the performance reported in the literature.

We have also downloaded the released model of Xu2020 (<https://github.com/j3xugit/RaptorX-3DModeling>) and evaluated it using identical MSA to ProFOLD. The results are summarized as follows.

Precision of the inter-residue contact predictions by Xu2020 and ProFOLD over CASP13 targets. Here, the precision for the most probable L , $L/2$ and $L/5$ long-range contacts over CASP13 targets are shown. Following the CASP13 criteria, T0953s1 is excluded as it has no long-range contacts

Method	FM (31 domains)			FM/TBM (12 domains)			TBM (61 domains)		
	L	$L/2$	$L/5$	L	$L/2$	$L/5$	L	$L/2$	$L/5$
Xu2020	0.537	0.689	0.801	0.583	0.758	0.902	0.634	0.773	0.878
ProFOLD	0.567	0.713	0.840	0.646	0.791	0.913	0.717	0.860	0.948

As shown in the above table, ProFOLD achieved higher precision for long-range contact prediction than Xu2020. (The difference between our evaluation and that reported by the original paper can be attributed to the different metrics used by our evaluation approach and RaptorX)

As the manuscript Xu2020 was submitted to bioRxiv after the submission of this manuscript and it is still under review, we did not include these comparison results in the main text.

Reviewers' Comments:

Reviewer #1:

Remarks to the Author:

I appreciate the more experimental results provided by the authors, which now show some advantage of the proposed 1d convolutional network for feature learning. However, I don't understand why the authors did not include the comparison with the updated RaptorX results into the main text. To my understanding, this comparison may reflect the real advantage of the learned features over manually-curated features since both RaptorX and ProFold are well-developed by their respective authors, not to mention that ProFold used 72 residual blocks while RaptorX used only 50 residual blocks.

P.S. Although not very important, I am a little puzzled by the fact that ProFold has a larger advantage over RaptorX on easier CASP13 targets than on harder targets. Nevertheless, on the CASP14 easier targets (i.e., TBM/FM targets) ProFold is less than 1% better than RaptorX in terms of contact precision.

COMMENTS

Reviewer #1 (Remarks to the Author):

I appreciate the more experimental results provided by the authors, which now show some advantage of the proposed 1d convolutional network for feature learning. However, I don't understand why the authors did not include the comparison with the updated RaptorX results into the main text. To my understanding, this comparison may reflect the real advantage of the learned features over manually-curated features since both RaptorX and ProFold are well-developed by their respective authors, not to mention that ProFold used 72 residual blocks while RaptorX used only 50 residual blocks.

We were pleased that the Reviewer #1 approved the experiments for showing the advantages of CopulaNet on feature learning. Following the reviewer's suggestion, we have now inserted the results as an extended table (Supplementary Table 1) and also expanded the description in Supplementary text.

The extended Supplementary Table 1 is shown below for the sake of quick reference.

Table 1 | Precision of the inter-residue contact predictions by RaptorX, A7D, trRosetta, and ProFOLD over CASP13 targets. Here, the precision for the most probable L , $L/2$ and $L/5$ long-range contacts over CASP13 targets are shown. Following the CASP13 criteria, T0953s1 is excluded as it has no long-range contacts

Method	FM (31 domains)			FM/TBM (12 domains)			TBM (61 domains)		
	L	$L/2$	$L/5$	L	$L/2$	$L/5$	L	$L/2$	$L/5$
RaptorX	0.439	0.562	0.686	0.530	0.645	0.807	0.652	0.798	0.899
A7D	0.470	0.596	0.712	0.584	0.731	0.866	0.694	0.826	0.901
trRosetta	0.489	0.627	0.750	0.576	0.728	0.867	0.649	0.792	0.917
Xu2020	0.537	0.689	0.801	0.583	0.758	0.902	0.634	0.773	0.878
ProFOLD	0.567	0.713	0.840	0.646	0.791	0.913	0.717	0.860	0.948

P.S. Although not very important, I am a little puzzled by the fact that ProFold has a larger advantage over RaptorX on easier CASP13 targets than on harder targets. Nevertheless, on the CASP14 easier targets (i.e., TBM/FM targets) ProFold is less than 1% better than RaptorX in terms of contact precision.

We understand the reviewer's concerns on the performance difference over easier targets and hard targets. As shown in the new Supplementary Table 1, when using

AlphaFold (A7D) as reference, Xu2020 performed better over hard targets (FM domains) but worse over easier targets (61 TBM domains and 12 FM/TBM domains). Similar results could be also observed for trRosetta’s performance. One plausible explanation might be that both Xu2020 and trRosetta focused on hard targets and were finely-tuned to make their prediction results much more accurate over hard targets than easy targets.

We also compared our method with Xu2020 over CASP14 targets. In particular, we downloaded the released models by Xu2020 (<https://github.com/j3xugit/RaptorX-3DModeling>) and evaluated them using identical MSA to ProFOLD. The performance of Xu2020 and ProFOLD are summarized as follows.

Table 1: Precision of the inter-residue contact predictions by Xu2020 and ProFOLD over CASP14 targets. Here, the precision for the most probable L , $L/2$ and $L/5$ long-range contacts over CASP14 targets are shown.

Method	Sequence databases	FM (22 domains)			FM/TBM (14 domains)		
		L	$L/2$	$L/5$	L	$L/2$	$L/5$
Xu2020	UniClust30 + UniRef90 + Metaclust	0.255	0.337	0.456	0.511	0.634	0.709
ProFOLD	UniClust30 + UniRef90 + Metaclust	0.295	0.394	0.518	0.566	0.703	0.779
Xu2020	BFD30 + UniRef90 + Metaclust	0.323	0.416	0.507	0.562	0.701	0.777
ProFOLD	BFD30 + UniRef90 + Metaclust	0.372	0.491	0.623	0.588	0.756	0.880

The table suggested that both Xu2020 and ProFOLD performed not very well when using the MSA generated by running DeepMSA (with default parameters) over a combined sequence database containing UniClust30, UniRef90 and Metaclust. In contrast, when replacing UniClust30 with a larger sequence databases BFD30, both Xu2020 and ProFOLD achieved significant performance improvement. In this case, ProFOLD is about 11% better than Xu2020 in terms of contact prediction precision (ProFOLD: 0.623 vs. Xu2020: 0.507 for the most $L/5$ probable contacts over 22 FM domains, and ProFOLD: 0.880 vs. Xu2020: 0.777 for the most $L/5$ probable contacts over 14 FM/TBM domains). These results clearly demonstrated the relative advantages of ProFOLD and the importance of generating high-quality MSAs. We are still working on improving MSA using more effective techniques and larger sequence databases (e.g. MGnify and JGI).